# OpenReview forum: "A Realistic Threat Model for Large Language Model Jailbreaks"
_ICLR.cc/2025/Conference — Submitted to ICLR 2025_

### Official Review · Reviewer_M1dJ · 2024-10-28

**Soundness:** 4
**Presentation:** 4
**Contribution:** 3
**Rating:** 8
**Confidence:** 4

**Summary:**

This paper introduces a unified threat model for evaluating jailbreak attacks on safety-tuned LLMs. Recognizing the multitude of existing jailbreak methods that vary in success rates, fluency, and computational effort, the authors propose a framework that combines constraints on perplexity—measuring deviation from natural text—and computational budget quantified by FLOPs. To achieve an LLM-agnostic and interpretable evaluation, they construct an N-gram model based on one trillion tokens. Adapting popular attacks within this new threat model, the paper benchmarks these methods against modern safety-tuned models on equal footing. The findings indicate that attack success rates are lower than previously reported, with discrete optimization-based attacks outperforming recent LLM-based methods. Furthermore, effective attacks tend to exploit infrequent N-grams, selecting sequences that are either absent from real-world text or rare, such as those specific to code datasets.

**Strengths:**

* Unified Threat Model: The paper addresses the critical need for a standardized framework to compare various jailbreak attacks, providing clarity in a field crowded with disparate methods.

* Interpretability: By employing an N-gram model for perplexity measurement, the threat model remains interpretable and LLM-agnostic, facilitating a deeper understanding of why certain attacks succeed or fail.

* Comprehensive Benchmarking: Adapting and evaluating popular attacks within the proposed threat model allows for a fair and rigorous comparison, advancing the discourse on LLM vulnerabilities.

**Weaknesses:**

* Comparison with Existing Methods: The paper would benefit from a direct comparison with existing perplexity detectors, such as the one proposed by Alon et al. (arXiv:2308.14132). This would contextualize the proposed model within the current state-of-the-art and highlight its relative advantages.

* Perplexity Measurement Limitations: While the N-gram model offers interpretability, it may not capture the nuances of natural language as effectively as model-based perplexity measures, potentially affecting the evaluation's accuracy.

**Questions:**

* Clarify Equation 1: Provide a complete and formal definition of the Judge function. This will enhance the paper's clarity and reproducibility.

* Enhance Comparative Analysis: Include a comparison with existing perplexity detectors, specifically the method proposed in arXiv:2308.14132.

---

> ### Author Response · Authors · 2024-11-21
> **Rebuttal by Authors**
>
> Dear reviewer M1dj,
>
> Thank you for your time and appreciation. We’re happy you agree that the “paper addresses the critical need for a standardized framework to compare various jailbreak attacks” and that “the threat model remains interpretable and LLM-agnostic, facilitating a deeper understanding of why certain attacks succeed or fail.” Moreover, we are grateful for your acknowledgment that “adapting and evaluating popular attacks within the proposed threat model allows for a fair and rigorous comparison.”
>
> Please find your questions addressed in the points below:
>
> ---
>
> **“The paper would benefit from a direct comparison with existing perplexity detectors, such as the one proposed by Alon et al. (arXiv:2308.14132)”**
>
> This is a good point that we had not formally included as data in the paper. To remedy this, we have now rerun our entire pipeline with an LLM-based perplexity filter (using the perplexity of the Llama-2-7b model that we use as default in the paper as a filter). We again carefully calibrate the TPR of this new filter and run strong adaptive attacks. The results can be seen in the following table:
>
> |                            | **ASR w/o LG2**                     | **ASR w/ LG2 (Input)**        | **ASR w/ LG2 (Input+Output)** |
> |----------------------------|-----------------------------------|---------------------------|---------------------------|
> | **Baseline**               | 0.68                             | 0.28                      | 0.20                      |
> | **Adaptive to N-gram LM PPL (TPR=99.9%)** | 0.46             | 0.32                      | 0.22                      |
> | **Adaptive to Self-PPL (TPR=99.9%)**      | 0.44             | 0.30                      | 0.18                      |
> | **Adaptive to Llama Guard 2**             | 0.46             | 0.46                      | 0.24                      |
>
> Here we see that the successful adaptation of the (blackbox, high perplexity) PRS attack leads to negligible difference in ASR of attacks against this new LLM filter (0.44) vs the old N-gram-based filter (0.46). We have added these results to the Appendix. We hope these results provide a clear datapoint that using the N-gram model does not compromise utility. This allows the other  important advantages of N-gram-based models for the purpose of our threat model to shine, namely that it is model-agnostic, easy to evaluate and interpretable (see Section 1 for details).
>
> ---
>
> **“ While the N-gram model offers interpretability, it may not capture the nuances of natural language as effectively as model-based perplexity measures”**
>
> What we find especially interesting here is that we ablated the capability of the N-gram model to capture nuances in language (via increasing the N, or via the comparison to LLM-based models as above), but we did not find an improvement in robustness against strong adaptive attacks. It seems like the 2-gram model is optimal in the sense that it models language well enough, but is robust enough not to increase the attack surface.
>
> ---
>
> **“Clarify Equation 1: Provide a complete and formal definition of the Judge function. ”**
>
> Thank you for your suggestion. We have introduced additional details in the description of Eq. 1. For more technical details of the judge function, please see App. B.
>
> ---
>
> Thank you for your feedback! We hope that we could answer all remaining questions. Please let us know if any follow-up questions arise.

---

> > ### Comment · Reviewer_M1dJ · 2024-11-25
> >
> > Thank you for your detailed rebuttal and the clarifications you provided. I appreciate the effort you have put into addressing my concerns and making the necessary changes. I have carefully reviewed your responses, and they have helped improve my understanding of your work.

---

### Official Review · Reviewer_PGG1 · 2024-11-02

**Soundness:** 3
**Presentation:** 3
**Contribution:** 2
**Rating:** 3
**Confidence:** 4

**Summary:**

The paper introduces a new "threat model" for LLM jailbreaks (I am not convinced that this is the right framing here) using an N-gram perplexity approach. There are some interesting ideas, but I think the framing needs adjustment before being ready for publication.

**Strengths:**

- I thought the idea of using the N-gram was interesting.
- The paper is fairly clear written, and the plots are clearly presented.
- I thought the analysis showing an N-gram perplexity constraint increased compute time for GCG interesting, and that it reduces ASR. The analysis comparing ASR against flops was generally very interesting.

**Weaknesses:**

- I'm not sure if the contribution is the threat model or the N-gram language model perplexity filter?
    - As far as I understand, the "threat model" is basically assuming the N-gram approach is the right way of doing things, but I am not sure that is clearly established here? If the point of the paper is to establish the threat model, there should be lots of evidence it is an appropriate defence.
    - I don't find this evidence in the paper. It is simply assumed that this is an appropriate defence?
- I am not convinced the threat model is the best one. I think the best threat model is trying to break what Frontier AI labs have released. I think claiming the threat model here is realistic is significantly overclaiming.
- I think the benefit and results section would benefit from making clear the implications of the results much more.

**Questions:**

- I am generally confused about the "threat model" framing, e.g., "universal threat model for comparing attacks across different models", how does the threat model actually allow you to compare?
- The claim in Table 1 is "The Lack of an LLM-Agnostic Threat Model Renders Attacks Incomparable." But I think the attacks are comparable, just on different axises? I could not follow precisely what the claim is here? I also think the most important thing is ASR? The paper makes the claim many times that the attacks are incomparable, but I just cannot follow this.
    - In terms of needed progress, the threat model is basically what frontier AI labs release? I'm not sure a new threat model is not what is needed.
- How does the N-gram model do on longer user queries? Presumably the perplexity increases substantially with longer queries. Does this mean that this defence would not work well with long-context models? Some of the latest models from Frontier labs can have very long context lengths. This makes me think the threat model might not actually be appropriate.

---

> ### Author Response · Authors · 2024-11-21
> **Rebuttal by Authors [1 / 2]**
>
> Dear reviewer PGG1,
>
> Thank you very much for your review. We’re glad you found our idea interesting, our paper fairly clearly written, the plots clearly presented, and the analysis of our threat model very interesting. Please find a point-by-point response below.
>
> -------
>
> **“I am generally confused about the "threat model" framing, e.g., "universal threat model for comparing attacks across different models,...”**
>
> * For us, a threat model answers the question: What is a minimal constraint (as additional defense layers/constraints would monotonically reduce utility) that ensures that the input to an LLM is likely to be normal user input? An ultimate judge of this would be a human, but clearly, this is not an option, so we need a proxy for this, which one can evaluate easily and independently of the employed LLM. Our window N-gram perplexity fulfills these requirements and measures the likelihood that a given prompt occurs in real, normal text, and we adjust the threshold such that 99.9% of all benign inputs would pass the filter so it is a minimal restriction not affecting the utility of the LLM.
>
> * A threat model restricts the set of inputs and thus can be seen as a weak version of a defense. Additionally, the budget of an attack is an often neglected parameter, and thus, we explicitly include it in our threat model. Our threat model is “universal” in the sense that it is independent of the LLM.
>
> ------
>
> **“I'm not sure if the contribution is the threat model or the N-gram language model perplexity filter?”**
>
> * Our contribution is the principled threat model we propose and evaluate as defined using the N-gram model perplexity.
> We chose perplexity filtering as our motivation and the starting point because, as we hypothesized and confirmed in Fig. 1 and Fig. 12, higher perplexity attacks lead to higher ASR, making existing attacks that do not control for this incomparable. We provide a principled construction of the threat model and an extensive evaluation of strong adaptive attacks. This combination provides us new insights into the strengths and weaknesses of each of the tested attacks and, for the first time, makes them comparable.
>
> -----
>
> **“As far as I understand, the "threat model" is basically assuming the N-gram approach is the right way of doing things, but I am not sure that is clearly established here? If the point of the paper is to establish the threat model, there should be lots of evidence it is an appropriate defence”**
>
> To model language, an N-gram-based language is clearly not the best way of doing things. LLMs do a better job on this. But that does not mean that, e.g., LLM-based perplexity is a better measure to filter out adversarial inputs, as LLM models offer a larger attack surface that an adversary can exploit.
>
> Thus, it is fair to ask how model-based perplexity compares as a filter. Note, however, that in Table 2, the attack is not adapted against Llama2-7b. To answer the question of how an N-gram LM vs LLM-based perplexity filter perform side-by-side, we reran our pipeline using Llama-2-7b as a filter, which, as for the N-gram, we calibrate so that a TPR of 99.9% is reached on benign prompts. We show the accuracy of principled adaptive attacks against both filters in the following table for the first 50 prompts of the benchmark:
>
> |                            | **ASR w/o LG2**                     | **ASR w/ LG2 (Input)**        | **ASR w/ LG2 (Input+Output)** |
> |----------------------------|-----------------------------------|---------------------------|---------------------------|
> | **Baseline**               | 0.68                             | 0.28                      | 0.20                      |
> | **Adaptive to N-gram LM PPL (TPR=99.9%)** | 0.46             | 0.32                      | 0.22                      |
> | **Adaptive to Self-PPL (TPR=99.9%)**      | 0.44             | 0.30                      | 0.18                      |
> | **Adaptive to Llama Guard 2**             | 0.46             | 0.46                      | 0.24                      |
>
> Based on these results, we see that both filters have a similar effect. Thus, in terms of ASR, there is no advantage to using an LLM-based filter. However, there are three important advantages of N-gram over LLM-based perplexity (see Section 1): i) it is model-agnostic; ii) it has negligible compute cost; iii) it is interpretable and provides insights into LLMs failure modes. But this is a good question, and we have added this table and an extended discussion to our revised version; see Table 6 and Fig. 14 in the Appendix.

---

> > ### Author Response · Authors · 2024-11-21
> > **Rebuttal by Authors [2 / 2]**
> >
> > **“I am not convinced the threat model is the best one. I think the best threat model is trying to break what Frontier AI labs have released. I think claiming the threat model here is realistic is significantly overclaiming.”**
> >
> > * While we do appreciate the sentiment, a potentially good analogy here is the difference between security research and hacking.
> > Companies such as OpenAI and Anthropic do not disclose the details of the potential threat models they use, which may include different environments, varying system prompts, and output and input-level defenses. Moreover, these environments are subject to change at the providers' discretion, effectively creating a set of threat models each bound to a specific point in time, which makes comparing attacks and achieving scientific progress challenging. Therefore, to propose attacks against these models, one must make assumptions about what constitutes the most "realistic threat model" and compare adaptive attacks within that context.
> >
> > * We do agree that good attacks in the proposed threat model should also transfer to the Frontier AI lab models. This is indeed a case, as we show in Figure 13, that a simple transfer of our generated prompts from Llama2-7B from PRS and GCG achieves high ASR against GPT 3.5, GPT-4, and GPT-4o, partially much higher ASR than what has been achieved for the source model Llama2-7B.
> >
> > * Taking this together indicates that our setup tracks progress on frontier model attacks.
> >
> > ----
> >
> > **“I think the benefit and results section would benefit from making clear the implications of the results much more.”**
> >
> > * In short, we propose a threat model and experimentally confirm that i) it makes a fair comparison of both black- and white-box attacks possible, ii) even for the strong safety-tuned models, iii) it considers the safety-utility trade-off, iv) it is model-agnostic, v) it has negligible compute cost, and vi) is interpretable and simple. To incorporate your suggestion, we add Section 5.4 to clarify the implications of our results.
> >
> > ------
> >
> > **“How does the threat model actually allow you to compare?”**
> >
> > * As discussed in Table 1 and Section 2, different attacks have wildly different assumptions. Two of the assumptions are the most critical according to our experiments and previous research: i) perplexity budget (jailbreaks with higher perplexity budget allowing for higher ASR) and ii) FLOP budget (higher FLOP budget allowing for higher ASR). The first part of our paper is about clearly establishing this.
> >
> > * Our threat model addresses both of these budget assumptions and ensures that each attack has the same budget, thus allowing us to compare the attacks and models.
> >
> > * The last part of our paper is about an even more fine-grained comparison of attacks and models because the N-gram-based filter is interpretable and allows the attribution of attacks to the corresponding type of training datasets.
> >
> > -----
> >
> > **“The claim in Table 1 is "The Lack of an LLM-Agnostic Threat Model Renders Attacks Incomparable." But I think the attacks are comparable, just on different axises? [...]  I also think the most important thing is ASR?”**
> >
> > ASR is indeed the most important metric. But if attack A implicitly assumes a 1000x budget compared to attack B on at least one of the axes, then one cannot fairly compare the achieved ASR using A and B. First, one has to i) define the budget that considers implicit assumptions (our threat model); ii) constrain it to the same values on each axis; iii) make a fair comparison in it (adaptive attacks) for both black- and white-box attacks; iv) consider the safety-utility trade-off (threshold selected correctly via TPR).
> >
> > -----
> >
> > **“I also think the most important thing is ASR? … In terms of needed progress, the threat model is basically what frontier AI labs release? I'm not sure a new threat model is not what is needed.”**
> >
> > * We are not sure if we get the question right. We show in Fig. 13 that a simple transfer attack can break GPT 3.5, GPT4, and even GPT4o with our adaptive PRS attack (best transfer at 99.9% TPR) with ASR even higher than for the source model Llama2-7B. In that sense, frontier LLMs are more vulnerable than less performant models. Given that the adaptive PRS attack in our threat model transfers better to GPT than the unrestricted PRS attacks, it shows that our threat model is very valuable in detecting security issues in frontier LLMs.
> >
> > * If this does not answer the question, we would be grateful if the reviewer could clarify the question.
> >
> > -----
> >
> > **“How does the N-gram model do on longer user queries? Presumably, the perplexity increases substantially with longer queries.”**
> >
> > We also wondered how our window-based measure would perform on real queries exceeding its window size 8. As you can see from Fig. 2a, the performance on realistic prompts from AlpacaEval is similar to what we would expect from window-based rejection.
> >
> > ------
> >
> > Thank you for your feedback. We hope we've addressed your questions.

---

> > > ### Comment · Reviewer_PGG1 · 2024-11-21
> > >
> > > Thanks for the response.
> > >
> > > > a threat model answers the question: What is a minimal constraint (as additional defense layers/constraints would monotonically reduce utility) that ensures that the input to an LLM is likely to be normal user input?
> > >
> > > I think your approach is basically placing additional constraints on the defender, but frontier labs don't implement a perplexity check. GCG attacks work. I think if you are arguing they should do this, you need evidence. I disagree strongly with the threat model framing, and further, I think you need evidence to show it is realistic. e.g., "how" is it "principled"
> > >
> > > >which may include different environments, varying system prompts, and output and input-level defenses
> > >
> > > actually, anthropic does release some information, see: https://www.anthropic.com/rsp-updates
> > >
> > > Overall, I cannot vote for acceptance here.

---

> > > > ### Author Response · Authors · 2024-11-22
> > > > **Rebuttal by Authors**
> > > >
> > > > Dear reviewer PGG1,
> > > >
> > > > Thank you for your prompt response. We appreciate your engagement with our work and would like to address the points you raised.
> > > >
> > > > ---
> > > >
> > > > What makes our approach “principled” is that we base it on the utility-safety trade-off: We choose the filter based on the rate of accepting benign prompts, which is very high for our proposed filter (99.9%), so that the utility of the LLM is almost unaffected. This way, we are placing constraints on **the attacker**. Further, we provide evidence that the benefits on the defending side (i.e., making it harder to break safety, see, e.g., Figure 1) outweigh the downsides (cost of a table lookup and 0.1% rejection rate for natural prompts).
> > > >
> > > > ---
> > > >
> > > > Thank you for the reference to Anthropic’s recent safeguards [1]. Similar to our setup, their pipeline includes a binary classification-based filter, formalizing part of what could be termed the “Anthropic threat model.” However, while the exact design of their classifier is not provided, this is actually a similar design to our threat model: We simply choose the most straightforward classifier, which is classifying un-natural text as not passing, based on perplexity.
> > > >
> > > > We further argue that perplexity should play a role in defense pipelines because, as you pointed out, GCG attacks are effective. Our results (Table 4, Figures 4 and 12) prove your point and provide clear evidence that the perplexity of inputs is positively correlated with ASR. Our results directly motivate the use of perplexity filtering, as is best seen in Figure 5, where tighter input constraints directly reduce attack success rates and increase attack costs.
> > > >
> > > > Other companies, such as NVIDIA also directly adopt perplexity-based filtering in their Guardrails framework [2], further supporting that this is a sensible setup.
> > > >
> > > > ---
> > > >
> > > > We do see that the “realistic threat model” framing might be potentially confusing. We originally termed the threat model “realistic” in reference to the increased realism of input queries and not to the relation to current guardrails used for frontier AI models. We are happy to iterate on this framing, or rephrase the title to come to a better compromise here. For example, a “Perplexity-based threat model”?
> > > >
> > > > ---
> > > >
> > > > We further agree that “threat model” might sound unconventional in comparison to framing in Frontier AI lab reports, e.g., Anthropic and NVIDIA [1, 2] do refer to the components of their system as “development safeguards.” We use “threat model” framing as, in our case, it describes the minimal desiderata and is consistent with established security literature.
> > > >
> > > > ---
> > > >
> > > > We understand that we might have a different perspective regarding the framing of this work. But, we hope we can find common ground regarding the value of our detailed evaluation of a perplexity-based threat model for a number of modern jailbreak attacks and novel adaptive attacks, providing value to the community.
> > > >
> > > > ----
> > > > **References**
> > > >
> > > > [1] https://www.anthropic.com/rsp-updates
> > > >
> > > > [2] https://docs.nvidia.com/nemo/guardrails/user_guides/jailbreak_detection_heuristics/README.html

---

### Official Review · Reviewer_YgJw · 2024-11-03

**Soundness:** 3
**Presentation:** 3
**Contribution:** 2
**Rating:** 5
**Confidence:** 4

**Summary:**

Jailbreaking attacks on large language models (LLMs) are widely studied in the literature. But those attack prompts are usually non-natural text. This paper proposes an N-gram method to measure the fluency of generated attack prompts. It shows that this simple approach can filter out several existing jailbreaking attacks and significantly reduce their attack success rates. The paper then proposes an adaptive attack which considers the N-gram of the attack prompt during generation. The results show it can boost the attack performance of existing jailbreaking methods.

**Strengths:**

1. A systematic study on jailbreaking attacks is an important direction and can lay the foundation for future research. This paper provides a good study in this space, which helps the community to better develop new techniques in attacking and protecting LLMs.
2. The proposed N-gram model is effective in filtering several attacks that generate jiberesh text. Those jailbreaking prompts are quite different from natural sentences, causing them easily detectable and hence not robust.
3. The evaluation is comprehensive, including multiple recent LLMs and safely aligned models. The baseline attacks are chosen from the state-of-the-arts.

**Weaknesses:**

1. It is known that several existing jailbreaking attacks generate non-natural text. There have been many proposed methods for filtering such jailbreaking prompts [1][2]. This insight mentioned in the paper is not new. The proposed approach of using the N-gram is straightforward. The novelty hence seems limited.
2. According to Table 2, using the perplexity measure of llama2-7b can distinguish the two optimized suffixes. Why is this approach not used to evaluate jailbreaking attacks in Tables 3 and 4? Additionally, there are many other filtering methods such as [1][2]. It is important to compare the performance of the proposed approach with these techniques.
3. The paper introduces an adaptive attack that considers the N-gram measure during adversarial prompt generation. It is strange that the paper introduces an attack against the proposed measure and then uses the same measure to evaluate the performance. It is similar to self-verifying correctness. It is suggested to use other filtering methods such as [1][2] and the llama perplexity to evaluate the final attack performance.
4. The case shown in Table 2 for the adaptive attack does not seem like natural text as well. Why does the N-gram model not filter this attack prompt? For example, the phrase “A questions their She” very unlikely exists in normal text. With the 8-gram model used in the paper, it should be able to filter out this. Could the authors explain why this case bypass the detection?


[1] Alon, Gabriel, and Michael Kamfonas. "Detecting language model attacks with perplexity." arXiv preprint arXiv:2308.14132 (2023).
[2] Inan, Hakan, et al. "Llama guard: Llm-based input-output safeguard for human-ai conversations." arXiv preprint arXiv:2312.06674 (2023).

**Questions:**

Please see the weaknesses section.

---

> ### Author Response · Authors · 2024-11-21
> **Rebuttal by Authors [1 / 3]**
>
> Dear reviewer YgJw,
>
> Thanks a lot for your review. We’re glad you found our N-gram model effective and our evaluation comprehensive and valuable for advancing new techniques in attacking and protecting LLMs. Please find answers to your questions below.
>
> ----
> **“There have been many proposed methods for filtering such jailbreaking prompts [1][2]. This insight mentioned in the paper is not new. The proposed approach of using the N-gram is straightforward. The novelty hence seems limited.”**
>
> Perplexity filters are indeed a sensible strategy to mitigate jailbreaks and have also been proposed in prior work [1,3], which we discuss in our related work. However, we do think that previous work (both references were rejected at ICLR 2024) has not done perplexity filters well. These works evaluate LLM-based perplexity filters but consider only weak adaptive attacks, if any. Notably, they both come to the conclusion that high-perplexity attacks, such as PRS or GCG, do not work, motivating many follow-up works, such as AutoDAN or BEAST, that are constructed to be low-perplexity attacks.
>
> However, we correctly execute a window-based perplexity filter, calibrate its TPR correctly, and then design strong adaptive attacks. With this principled approach, we are able to provide an accurate assessment of the strength of perplexity filters, and we do find that attacks like PRS and GCG actually outperform newly-proposed low-perplexity attacks - which we think is a valuable contribution to the community and a strength of our benchmark design.
>
> On top of this, our approach has further advantages, as we discuss in the paper:
>   * We use a perplexity filter based on a bigram model agnostic of the LLM, which allows us to compare this filter across models.
>   * We discuss the utility-robustness tradeoff intensively and use a very conservative threshold in our threat model so that 99.9% of benign prompts pass the N-LM filter. Note that this threshold is agnostic of the employed LLM, and thus, our threat model transfers across LLMs - making it straightforward to test new attacks and models.
>   * We can interpret the potential reasons exploited by jailbreaks by analyzing the origin of the bigrams used in the attacks (see Section 5.4).
>
> Regarding Llama-Guard 2 [2], we have evaluated this filter against 50 prompts generated by PRS for Llama2-7B: 26 of 50 prompts pass. This leads to an ASR of 28% with input filtering and 20% with input-output filtering at 96% TPR [2], whereas applying our N-gram-based perplexity filter leads to 0% ASR at 99.9% TPR (see Table 3), meaning that even without adaptive attacks, the guard underperforms against the strong attacks we evaluate. Moreover, for the prompts generated by our adaptive PRS to our N-gram filter, even 33 out of 50 pass Llama-Guard 2, leading to an ASR of 32% for input filtering and an ASR of 22% for input-output filtering. We have now added the results for Llama-Guard 2 in Section I in the Appendix to clarify this point.
>
> Thus, we hope we can respectfully disagree here - we think that our adaptive attacks and principled approach of our threat model are novel and provide valuable insights for the community.

---

> > ### Author Response · Authors · 2024-11-21
> > **Rebuttal by Authors [2 / 3]**
> >
> > **“According to Table 2, using the perplexity measure of llama2-7b can distinguish the two optimized suffixes. Why is this approach not used to evaluate jailbreaking attacks in Tables 3 and 4? Additionally, there are many other filtering methods such as [1][2].”**
> >
> > It is a fair point to ask how model-based perplexity compares as a filter. Note, however, that in Table 2, the attack is not adapted against Llama2-7b. To answer the question of how an N-gram LM vs LLM-based perplexity filter perform side-by-side, we reran our pipeline using Llama-2-7b as a filter, which, as for the N-gram, we calibrate so that a TPR of 99.9% is reached on benign prompts. We show the accuracy of principled adaptive attacks against both filters in the following table for the first 50 prompts of the benchmark:
> >
> > |                            | **ASR w/o LG2**                     | **ASR w/ LG2 (Input)**        | **ASR w/ LG2 (Input+Output)** |
> > |----------------------------|-----------------------------------|---------------------------|---------------------------|
> > | **Baseline**               | 0.68                             | 0.28                      | 0.20                      |
> > | **Adaptive to N-gram LM PPL (TPR=99.9%)** | 0.46             | 0.32                      | 0.22                      |
> > | **Adaptive to Self-PPL (TPR=99.9%)**      | 0.44             | 0.30                      | 0.18                      |
> > | **Adaptive to Llama Guard 2**             | 0.46             | 0.46                      | 0.24                      |
> >
> > Based on these results, we see that both filters have a similar effect, so there is no advantage to using an LLM-based filter in terms of ASR. However, there are three important advantages of N-gram over LLM-based perplexity (see Section 1): i) it is model-agnostic; ii) it has negligible compute cost; iii) it is interpretable and provides insights into LLM failure modes. But this is a good question, and we have added this table and an extended discussion to our revised version; see Table 6 and Fig 14 in the Appendix.
> >
> > Regarding the Llama Guard 2 [2], as mentioned in the previous answer, Llama Guard 2 is not even efficient against the original PRS attack. In contrast, our N-gram-based perplexity filter would reject all these attacks. We have added these Llama Guard 2 results in Section I of the revised version's appendix.
> >
> > We would appreciate feedback if this resolves your concerns.
> >
> > ---
> >
> > **“The paper introduces an attack against the proposed measure and then uses the same measure to evaluate the performance. It is similar to self-verifying correctness.”**
> >
> > * This appears to be an important misunderstanding. We define a threat model for the attack in Section 3.3 that lays out the potential actions and knowledge of the attacker and defender. Using the knowledge described in the threat model, we then work to provide the best possible attack against the proposed measure.
> >
> > * This kind of adaptive attack evaluation is a foundational tenet of adversarial machine learning [4, 5]. This is the only way to correctly assess the robustness of the proposed threat model. Proposing a new threat model or defense and not executing this evaluation would provide a false sense of security and would have been a self-fulfilling prophecy.
> >
> > * In case your comment referred to the judge model used to evaluate the performance of the attack, note that this model (which scores attacks for harmfulness) is an independent model not available to the attacker or defender (as outlined in our threat model) and is used to calculate ASR. For us, this judge model is the well-established HarmBench evaluator.

---

> ### Author Response · Authors · 2024-11-21
> **Rebuttal by Authors [3 / 3]**
>
> **“The phrase “A questions their She” very unlikely exists in normal text. With the 8-gram model used in the paper, it should be able to filter out this. Could the authors explain why this case bypass the detection?”**
>
> * Please note that we measure mean 2-gram perplexity over a window size of length 8. We found this combination to be optimal in our ablations  (see Section 3.2 and App. C). This choice is optimized to separate benign and malicious prompts. We note that, e.g., extending the N-gram to an 8-gram would score this string as more surprising, but, due to the sparsity of the 8-gram, would require a higher threshold to achieve a 99.9% TPR on benign data, leading to a loss in utility.
>
> * For the 2-gram model, the evaluation becomes clear when investigating the tokenization of the string: “_A”, “_questions”, “_their”, “_She”. The corresponding bigram “A questions”  (1630 counts in Dolma), “questions their” (18803 counts), “their She” (8888 counts) all appear in natural text, e.g. as part of “... Q & A questions”, “He questions their…”, “their Sheets”. This observation highlights the advantage of an interpretable threat model, as for LLM-based perplexity, it would be difficult or even impossible to trace the origin of suffixes in the training data.
>
> ---
>
> **References**
>
> [1] Alon et al., “Detecting language model attacks with perplexity.”
>
> [2] Llama Guard 2, https://github.com/meta-llama/PurpleLlama/blob/main/Llama-Guard2/MODEL_CARD.md
>
> [3] Jain et al., “Baseline Defenses for Adversarial Attacks Against Aligned Language Models.”
>
> [4] Tramer et al., “On Adaptive Attacks to Adversarial Example Defenses.”
>
> [5] Carlini and Wagner, “Adversarial examples are not easily detected: Bypassing ten detection methods.”
>
> ---
>
> Thank you for your in-depth questions and comments. We hope to have addressed your concerns in our response above. Please let us know if you have any follow-up questions.

---

> > ### Comment · Reviewer_YgJw · 2024-11-25
> >
> > I appreciate the comprehensive response with additional results.
> >
> > After reading the rebuttal, the advantage of the proposed filtering method over LLM-based approaches seems to lie in its lower computational cost. However, I disagree with the model-agnostic argument, as LLM-based methods only analyze the text sequence and are also model-agnostic.
> >
> > Overall, I believe the paper has merit, but it may not meet the acceptance threshold for ICLR.

---

> > > ### Author Response · Authors · 2024-11-26
> > > **Rebuttal by Authors**
> > >
> > > Dear reviewer YgJw,
> > >
> > > Thank you for your response.
> > >
> > > * We would like to emphasize that the implications of our work extend far beyond the introduction of yet another perplexity filter. Previous works, including [1, 2], have not properly addressed adaptive attacks, which is a critical limitation. We, however, solve this limitation and show that our approach *generalizes well across different attacks.*
> > >
> > > * This limitation was pointed out at the review stage, and neither work [1, 2] was accepted at ICLR 2024. However, the false notion of perplexity filter integrity has significantly influenced the field, steering it towards less effective but more fluent attacks, such as PAIR, AutoDAN, and BEAST. Our approach allows us to compare attacks systematically and show for the first time that *discrete optimization attacks, such as GCG and PRS, significantly outperform more recent fluent attacks (PAIR, AutoDAN, and BEAST) in this fair setting.*
> > >
> > > * Our paper is the first to propose an adaptation method for discrete optimization attacks. This approach is effective against our own filter and, as demonstrated in Section I, also works against LLM-based filters. Thus, our approach *generalizes well across different filters.*
> > >
> > > * Additionally, we agree that LLM-based filters can be considered model-agnostic if a fixed model is employed to evaluate all attack queries. However, this adds *additional inference costs for the defender*. We intended to highlight with the “model-agnostic” that the N-gram filter does not rely on any specific model architecture and is built directly on the data.
> > >
> > > Overall, we thank you for your comments and questions. Let us know if you have additional questions. If not, we kindly ask you to consider raising the score.
> > >
> > > -----
> > >
> > > **References**
> > >
> > > [1] Alon et al., “Detecting language model attacks with perplexity.”
> > >
> > > [2] Jain et al., “Baseline Defenses for Adversarial Attacks Against Aligned Language Models.”

---

### Official Review · Reviewer_ATuy · 2024-11-04

**Soundness:** 2
**Presentation:** 3
**Contribution:** 2
**Rating:** 5
**Confidence:** 4

**Summary:**

This paper proposes a new threat model to compare different attacks. This threat model includes a perplexity filter based on an N-gram language model and constraints on FLOPs. A fine-tuned LLM judge measures the ASR. Many existing attacks fail in this threat model. With the consideration of the proposed perplexity filter, the adapted attacks can restore many ASRs.

**Strengths:**

1. A unified model to evaluate different attacks.
2. N-gram LM has certain advantages.

**Weaknesses:**

1. Only considered white-box attacks. In the real world, black-box attacks are more practical. As shown by Figure 13, white-box attacks in this new threat model have lower transferability. That is, this threat model cannot measure black-box attacks very well.

2. The perplexity filter is not new.

3. There are other defenses such as [instruction filter](https://arxiv.org/abs/2312.06674), and [random perturbation](https://arxiv.org/abs/2310.03684), etc. Why doesn't the threat model consider them?

4. Evidence is needed to show that N-gram LM is better than LLM. Some experiments are necessary.

**Questions:**

Please see the weaknesses.

---

> ### Author Response · Authors · 2024-11-21
> **Rebuttal by Authors [1 / 2]**
>
> Dear reviewer ATuy,
>
> Thank you for your review and questions. We are encouraged that you appreciate that we introduce the unified threat model and acknowledge the advantages of N-gram LM. Please find answers to your questions below.
>
> -----
>
> **“Only considered white-box attacks. In the real world, black-box attacks are more practical.”**
>
>  *  Our threat model indeed allows white-box access to the model. We think this is the most principled way of setting up the threat model and estimating the worst-case attack success rates against these models, especially considering the potential of transfer from (potentially very similar) open-source models [3]. *However*, within this threat model, we evaluate attacks from the entire spectrum: PAIR is a fully black-box attack, PRS and BEAST are score-based black-box attacks, and GCG uses the full white-box model. We have clarified this in Table 1.
>
> -------
> **“As shown by Figure 13, white-box attacks in this new threat model have lower transferability. That is, this threat model cannot measure black-box attacks very well.”**
>
>   * This is potentially a misunderstanding. In Fig.13, we report the attack success rate of all generated prompts we generate with PRS on Llama-2-7B against black-box models - we do not report the transfer rate  (the ratio of the number of successful attacks against the target transfer model to the successful ones against the source Llama-2-7B model). The ASR of the attack against the source model, Llama-2-7B, decreases as we tighten the perplexity constraint, yet the ASR on the target model stays mostly constant. As such, the transfer rate actually increases!
> * We agree that this is hard to parse out of Fig.13, and so we have modified the figure to now overlay the original ASR on the source model in red. To further validate our transfer results, we added new results for Llama 3.1-405B models and a WizardLM model, all of which follow the same trend of mostly constant ASR on the target model while ASR on the source model decreases.
>
> * Overall, we think it is an interesting validation of our benchmark that the best attack in our evaluation is a score-based black-box attack that also transfers well.
>
> ----
>
> **Regarding the novelty of perplexity filters**
>
> * Perplexity filters are indeed a sensible strategy to mitigate jailbreaks and have also been proposed in prior work [1,2], which we discuss in our related work. However, we do think that previous work (both references were rejected at ICLR 2024) has not done perplexity filters well. These works evaluate LLM-based perplexity filters but consider only weak adaptive attacks¹, if any. Notably, they both come to the conclusion that high-perplexity attacks, such as PRS or GCG, do not work, motivating many follow-up works, such as AutoDAN or BEAST, that are constructed to be low-perplexity attacks.
> However, we correctly execute a window-based perplexity filter, calibrate its TPR correctly, and then design strong adaptive attacks. With this principled approach, we are able to provide an accurate assessment of the strength of perplexity filters, and we do find that attacks like PRS and GCG actually outperform newly-proposed low-perplexity attacks - which we think is a valuable contribution to the community and a strength of our benchmark design.
>
> * On top of this, our approach has further advantages, as we discuss in the paper:
>   * We use a perplexity filter based on a bigram model agnostic of the LLM, which allows us to compare this filter across models
>   * We discuss the utility-robustness tradeoff intensively and use a very conservative threshold in our threat model so that 99.9% of benign prompts pass the N-LM filter. Note that this threshold is agnostic of the employed LLM, and thus, our threat model transfers across LLMs - making it straightforward to test new attacks and models.
>   * We can interpret the potential reasons exploited by jailbreaks by analyzing the origin of the bigrams used in the attacks (see Section 5.4).
>
> ¹ We construct as one of our major contributions strong adaptive versions of several well-established attacks against our bigram-based perplexity filter that work successfully against strong safety-tuned models. We want to stress that [1] using model-based perplexity did not consider adaptive attacks at all, and [2] constructed an adaptive attack for GCG but reported a low ASR even for Vicuna-7B at a TPR for benign text of 85%.

---

> ### Author Response · Authors · 2024-11-21
> **Rebuttal by Authors [2 / 2]**
>
> **“There are other defenses such as instruction filter, and random perturbation, etc. Why doesn't the threat model consider them?”**
>
> * It is important here to separate the general threat model we propose from any particular defense that could be employed inside such a threat model. To us, this is analogous to an $L_\infty$ constraint in adversarial attacks in vision. To achieve a sensible threat model containing a good model-agnostic constraint, we employ the N-gram perplexity with the setting we propose and carefully evaluate it in a principled way.
>
> * We agree that combining existing defenses, as mentioned by the reviewer, might offer an additional advantage when defending models; this is orthogonal to the current direction of the paper but an interesting topic for the future.
>
> * To provide additional data, we evaluated LLama-Guard 2 as a defense in our threat model.
> However, the instruction filter of Llama-Guard 2 is very weak: 26 of the 50 prompts of the original PRS attack for Llama2-7B pass the guard with input filtering at 96% TPR [4] and so already achieve an ASR of 28% with input filtering and 20% with input-output filtering (compared to the original ASR of 68%), whereas using our NLM-perplexity filter leads to an ASR of 0% at 99.9% TPR (see Table 3).
>
> -----
>
> **Evidence is needed to show that N-gram LM is better than LLM.**
>
>  We do think that there are a number of advantages to the setup we propose (as discussed above), but to address this question directly, we have now added new results using an LLM-based perplexity filter:
>
> |                            | **ASR w/o LG2**                     | **ASR w/ LG2 (Input)**        | **ASR w/ LG2 (Input+Output)** |
> |----------------------------|-----------------------------------|---------------------------|---------------------------|
> | **Baseline**               | 0.68                             | 0.28                      | 0.20                      |
> | **Adaptive to N-gram LM PPL (TPR=99.9%)** | 0.46             | 0.32                      | 0.22                      |
> | **Adaptive to Self-PPL (TPR=99.9%)**      | 0.44             | 0.30                      | 0.18                      |
> | **Adaptive to Llama Guard 2**             | 0.46             | 0.46                      | 0.24                      |
>
> * Here, we reran our entire pipeline using Llama2-7B to measure LLM-based perplexity and, additionally - using Llama Guard 2. For the Llama2-7B (Self-PPL) filter, we correctly calibrate it to a 99.9% TPR rate on benign prompts and run strong adaptive attacks against it using PRS. To our knowledge, adapting PRS to LLM-based defenses using rejection sampling is novel.
>
> * As shown in the table above, the ASR achieved with the N-gram-based filter is 46% versus 44% with the Llama2-7B-based PPL filter and 46% - with the Llama Guard 2. Based on these preliminary results, we see that N-gram-based and LLM-based filters have a similar effect so that in terms of ASR, there is no advantage to using the LLM-based filters. However, there are three important advantages of N-gram over LLM-based PPL (see Section 1) that make it much better suited for the design of our threat model: i) model-agnostic; ii) negligible compute cost; iii)  interpretable and provides insights to LLMs failure modes.
>
> ----
>
> **References**
>
> [1] Alon et al., “Detecting language model attacks with perplexity.”
>
> [2] Jain et al., “Baseline Defenses for Adversarial Attacks Against Aligned Language Models.”
>
> [3] Zou et al., “Universal and Transferable Adversarial Attacks on Aligned Language Models.”
>
> [4] Llama Guard 2, https://github.com/meta-llama/PurpleLlama/blob/main/Llama-Guard2/MODEL_CARD.md
>
> -----
>
> Overall, thank you for your questions. We think we have addressed your concerns, but please let us know if there are follow-up questions you would like us to address.

---

> ### Author Response · Authors · 2024-11-26
> **Rebuttal by Authors**
>
> Dear reviewer ATuy,
>
> Thank you once again for your suggestions on improving the paper, which we were happy to incorporate (see the latest revision).
>
> We would be glad to answer your questions, if you have any and in case we have answered all of your points, we would kindly ask you considering raising the score.

---

### Author Response · Authors · 2024-11-25
**Meta Reply by Authors**

We thank all reviewers for their time and assessment of our work. We appreciate that they found our evaluation “comprehensive” (YgJw, M1dJ), the paper “clearly written” (PGG1), and highlighted the "unified model to evaluate different attacks" as our strength (ATuy).

\
**Comparison with LLM-based filters**

First, we want to clarify the difference between the N-gram language model- and LLM-based filters and why we chose the former - a point that came up in several reviews (ATuy, YgJw, and M1dJ).

We do think that there are a number of advantages to the setup we propose (as discussed above), but to address this question directly, we have now added the suggested comparison to the LLM-based filters:

|                            | **ASR w/o LG2**                     | **ASR w/ LG2 (Input)**        | **ASR w/ LG2 (Input+Output)** |
|----------------------------|-----------------------------------|---------------------------|---------------------------|
| **Baseline**               | 0.68                             | 0.28                      | 0.20                      |
| **Adaptive to N-gram LM PPL (TPR=99.9%)** | 0.46             | 0.32                      | 0.22                      |
| **Adaptive to Self-PPL (TPR=99.9%)**      | 0.44             | 0.30                      | 0.18                      |
| **Adaptive to Llama Guard 2**             | 0.46             | 0.46                      | 0.24                      |

* Here, we reran our entire pipeline using Llama2-7B to measure LLM-based perplexity and, additionally - using Llama Guard 2. For the Llama2-7B (Self-PPL) filter, we calibrate it to a 99.9% TPR rate on benign prompts and run strong adaptive attacks against it using PRS. To our knowledge, adapting PRS to LLM-based defenses using rejection sampling is novel.

* As shown in the table above, the ASR achieved with the N-gram-based filter is 46% versus 44% with the Llama2-7B-based PPL filter and 46% - with the Llama Guard 2. Based on these results, we see that N-gram-based and LLM-based filters have a similar effect so that in terms of ASR, there is no advantage to using the LLM-based filters. However, there are three important advantages of N-gram over LLM-based filters (see Section 1) that make it much better suited for the design of our threat model: i) model-agnostic; ii) negligible compute cost; iii) interpretable and provides insights to LLMs failure modes.

-----

\
**Incorporated suggestions**

We also appreciate the constructive feedback from all reviewers. Thanks to their comments, we have improved the paper as follows:

* Reviewers (ATuy, YgJw, and M1dJ) suggested providing a comparison to the LLM-based perplexity.

  * To incorporate this, we added Section I in the Appendix and discussed it in answers to the reviewers below.

* Reviewers (ATuy, PGG1) suggested discussing the attack's performance when those are evaluated against closed-source models in more detail.

  * We expanded and improved Section H in the Appendix and discussed it in the answers below.

* Per the suggestion of Reviewer PGG1, we provided a clearer description of the implications of the results in Section 5.5.

* Additionally, we fixed typos and phrasing throughout the paper.
* Lastly, we understand that Reviewer PGG1 might have a different framing of our approach. Still, we hope we can find common ground regarding the value of our detailed evaluation of a perplexity-based threat model for a number of modern jailbreak attacks and novel adaptive attacks, providing value to the community.

  While it was pointed out that companies like OpenAI and Anthropic [1] do not use perplexity filters, our results clearly show that even after a stress test (using comprehensive, adaptive attacks), these filters do indeed reduce attack success and increase the attacker’s effort. As such, our work provides strong evidence that these filters should also be deployed by these companies, as already done by other companies such as NVIDIA [2].

-----

\
**References**

[1] https://www.anthropic.com/rsp-updates

[2] https://docs.nvidia.com/nemo/guardrails/user_guides/jailbreak_detection_heuristics/README.html

---

### Meta-Review · Area_Chair_5Ajq · 2024-12-16

**Metareview:**

The paper proposes a unified threat model for jailbreaks, that comprises attack success but also fluency and computational effort.

I think this paper uses the term "threat model" in a somewhat unusual and confusing way, by referring to it as a "weak version of a defense" (c.f. https://openreview.net/forum?id=1kMTJnqmyl&noteId=Dbv15jJwz2).
In security research, a threat model is typically a set of constraints or assumptions placed on an attacker (e.g., what are the attacker goals, what invariants would an attack violate, what can the attacker do or not do).

So a threat model might be: the attacker wants to jailbreak a model and has to do so in a way that bypasses a perplexity filter, and uses at most X computation.

But there are two issues I see with this:
(1) the threat model seems directly tied to a very specific way of doings perplexity filtering, namely with an N-gram model. Why not treat this more generally?
(2) placing bounds on an adversary's computational power is tricky, unless we go the cryptographic route and claim an attack is intractable. What if one attack is parallelizable and another isn't? What if one attack has been better optimized than another? Generally I would not make computation part of a threat model unless there's a good reason to claim that an adversary would be strongly computationally bounded.

Overall, while the study of perplexity filters is interesting, I would thus recommend rejection and encourage the authors to clarify what they mean by a "threat model" in this context.

**Additional Comments On Reviewer Discussion:**

The reviewer discussion did surface some of the problems discussed above (e.g., in the discussion with reviewer PGG1) but did not fully resolve them.

---

### Decision · Program_Chairs · 2025-01-22

Reject